

# A comparative survey of the impacts of extreme rainfall in two international case studies

Matthieu Spekkers[1], Viktor Rözer[2], Annegret Thieken[3], Marie-claire ten Veldhuis[1], and Heidi Kreibich[2]

[1]Delft University of Technology, Department of Water Management, Stevinweg 1, 2628 CN Delft, the Netherlands
[2]German Research Centre for Geosciences, Section 5.4 Hydrology, Telegrafenberg, 14473 Potsdam, Germany
[3]University of Potsdam, Institute of Earth and Environmental Science, Karl-Liebknecht-Strasse 24–25, 14476 Potsdam-Golm, Germany

*Correspondence to:* Viktor Rözer (vroezer@gfz-potsdam.de)

**Abstract.**

Flooding is assessed as the most important natural hazard in Europe, causing thousands of deaths, affecting millions of people and accounting for large economic losses in the past decade. Little is known about the damage processes associated with extreme rainfall in cities, due to a lack of accurate, comparable and consistent damage data. The objective of this study is to investigate the impacts of extreme rainfall on residential buildings and how affected households coped with these impacts in terms of precautionary and emergency actions. Analyses are based on a unique dataset of damage characteristics and a wide range of potential damage explaining variables at the household level, collected through computer-aided telephone interviews (CATI) and an online survey. Exploratory data analyses based on a total of 859 completed questionnaires in the cities of Münster (Germany) and Amsterdam (the Netherlands) revealed that the uptake of emergency measures is related to characteristics of the hazardous event. In case of high water levels, more efforts are made to reduce damage, while emergency response that aims to prevent damage is less likely to be effective. The difference in magnitude of the events in Münster and Amsterdam in terms of rainfall intensity and water depth, is probably also the most important cause for the differences between the cities in terms of the suffered financial losses. Factors that significantly contributed to damage in at least one of the case studies are water contamination, the presence of a basement in the building and people's awareness of the upcoming event. Moreover, this study confirms conclusions by previous studies that people's experience with damaging events positively correlates with precautionary behaviour. For improving future damage data acquisition, we recommend to include cell-phones in a CATI survey to avoid biased sampling towards certain age groups.

## 1 Introduction

More than 200 major flood events occurred in Europe between 1998 and 2009, causing 1126 deaths, displacement of about half a million people and around EUR 52 billion insured economic losses (European Environment Agency, 2010). These lumped statistics include various types of flooding, including fluvial floods, flash floods, as well as pluvial floods in urban areas that are triggered by extreme rain events overwhelming urban drainage systems. Currently, only little is known about the contributions of the different flood types and characteristic damage processes.





To better manage floods and to reduce their impacts, the European Union launched the Floods Directive in 2007 (European Commission, 2007). When implementing the directive, most of the countries concentrated on fluvial and coastal floods and neglected pluvial floods despite their damaging character (European Commission, 2015). In Copenhagen, for example, the pluvial flood of July 2011, which caused EUR 807 million of insured losses (Garne et al., 2013), demonstrated that the ad-

verse consequences of extreme rainfall must not be neglected. Damage due to extreme rainfall in cities is not only associated with pluvial flooding. A considerable amount of damage is caused by rainwater directly entering the building through roofs (Spekkers et al., 2015).

A prerequisite for an adequate management of the risks of extreme rainfall is a quantitative analysis of the hazard and its potential impacts. To quantify impacts, processes that govern damage caused by extreme rainfall have to be analysed,

understood and finally used to derive quantitative loss models. Accurate, comparable and consistent data on impacts of extreme rainfall and potentially influencing factors, gathered on the scale of flood-affected properties, serve as a good basis. While such comprehensive data sets have been collected for fluvial floods in recent years (e.g. Gissing and Blong, 2004; Thieken et al., 2005, 2016; Kreibich et al., 2007; Kienzler et al., 2015), data collection for extreme rainfall is rare and samples are much smaller (Rözer et al., 2016; Van Ootegem et al., 2015).

Two approaches to collect ex-post damage data can be distinguished. Large data sets originate from loss adjustments by insurers or from pay-outs of governmental disaster funds or other risk transfer schemes. Such data sets provide a complete picture of the losses of insured households and properties with regard to the total amount of losses and also their spatial as well as temporal distribution. However, these data do no contain information on damage conditions and processes underlying damage estimates. Therefore, they are only of limited use for loss model development (e.g. Spekkers et al., 2014). In addition,

loss data from risk transfer schemes, particularly from flood insurance, may be biased. Insurance data only cover households that are insured and thus not necessarily the whole affected population. Moreover, insurance contracts commonly include a deductible as well as an excess rate, i.e. the insured household has to cover small losses as well as losses which exceed the excess rate by it own. Thus, these costs have to be added to the pay-outs in order to receive the total loss (e.g. Thieken et al., 2006).

Scientific surveys can help to overcome some of the problems associated with insurance data sets. Surveys allow the collection of detailed information on the property scale including many factors that might influence the amount and type of damage, such as hazard characteristics at the affected property, characteristics of the affected structure including property-level precautionary and emergency measures as well as socio-economic variables of the affected households. However, due to the high costs and the dependence on the willingness of affected residents to participate in the survey, only a sample of the affected

population can be investigated and is hence covered by the data. In contrast to data from insurance, surveys are not necessarily restricted to residents that suffered from damage. In fact, residents that live in the affected area, but did not experience damage may contribute information that is important for damage analysis and risk mitigation (Van Ootegem et al., 2015).

In the past decades, several scientific surveys have been conducted in the aftermath of severe flood events, focussing on private households in Germany (e.g. Kreibich et al., 2005; Thieken et al., 2005, 2007; Kienzler et al., 2015; Thieken et al.,

2016). Only a few surveys have been carried out to investigate damage due to extreme rainfall. For example, Van Ootegem



et al. (2015, 2016) conducted in 2013 a mail survey among pluvial flood victims in Flanders, the northern part of Belgium. People were asked to report how much damage they suffered to several parts of the building as well as the building contents. Explanatory variables were collected, such as building characteristics, behavioural indicators and socio-economic variables, to construct multivariate damage models for pluvial floods. Rözer et al. (2016) used data collected through computer-aided

telephone interviews (CATI) to analyse three pluvial flood events in Germany. Rözer et al. (2016) found emergency response to play a bigger role in pluvial flood damage mitigation than for fluvial floods, because of the relative low water depths associated with pluvial floods and a low risk awareness among people for this type of flooding. Poussin et al. (2015) conducted a mail survey in three regions in France to investigate how households reacted in terms of mitigation measures for different types of flooding, including pluvial flooding. They found that the effectiveness of flood mitigation measures depends on the

characteristics of the flood hazard.

For this type of analysis, the risk management was found to be a valuable framework (Thieken et al., 2007; Kienzler et al., 2015; Rözer et al., 2016) . This cycle generally consists of three phases (see Fig. 1).

1. Response and recovery: Just before, during and immediately after a damaging event, residents take emergency measures to limit adverse effects of the event and start to clean-up and repair damage as soon as possible in order to regain the

pre-event standard of living;

2. Risk analysis and event assessment: In order to create a sound knowledge base for risk management, a phase of risk analysis and event assessment should be performed including the investigation of the adverse consequences;

3. Disaster risk reduction: In the face of a next disaster, residents plan and implement adequate precautionary and preparatory measures that aim at preventing and mitigating risks.

In this paper, we analyse the impacts of extreme rainfall to residential buildings in the cities of Münster (Germany) and Amsterdam (the Netherlands) as well as precautionary behaviour and emergency response by households, using the risk management cycle as an aid to analyse and present results. The two cities suffered from extreme rainfall in the past years, most notably the severe weather event of 28 July 2014 that caused rainfall damage in parts of northern and central Europe. Within the risk management cycle, we focused on the following research questions in particular:

1. How did residents in Amsterdam and Münster respond to a hazardous rain event by undertaking emergency measures?

2. What is the financial damage to building structure and building content due to a hazardous rain event?

3. How does the level of precaution and other possible explanatory variables affect the height of these losses?

4. How prepared are residents in Amsterdam and Münster for extreme rainfall?

5. Does experience with previous damaging rain events affect people's precautionary behaviour?

These questions were indicated as being important for flood risk management during panel discussions with professionals working for the city of Amsterdam. Similar questions were also discussed in related studies by Kienzler et al. (2015); Rözer et al. (2016).



Scientific surveys were administered among affected households in Münster and Amsterdam to collect information on self-reported financial losses caused by damage to building structure and building content, as well as factors potentially influencing damage, such as hazard, building and socio-economic characteristics. A questionnaire was developed for the purpose of investigating the impacts of intense local rainfall. It has a flexible structure and is set up in open source software to make it easily
adaptable and applicable to other cases.

After briefly describing the two case studies and the damage data collection campaign in the next section, we discuss the result of the case study comparison in Sect. 3. We then discuss possible methodological biases and differences between the case studies due to hazard and regional characteristics (Sect. 4). Conclusions are summarised in Sect. 5.

## 2  Data and methods

### 2.1  Case studies

Two case studies are central in this paper: the cities of Münster (Germany) and Amsterdam (the Netherlands). Both cities suffered rainfall damage caused by a synoptic weather event that occurred on 28 July 2014. The following two sections describe the case studies in detail. Key features of the two case studies are summarized in Table 1.

### 2.1.1  Amsterdam

On 28 July 2014, the city of Amsterdam (population: 830,000, area: 230 $\text{km}^2$) was hit by an extreme rainfall. Between 07:30 UTC and 14:00 UTC, a total of 93 mm of rainfall was accumulated in 6.5 hours, based on radar data from the Royal Netherlands Meteorological Institute (KNMI, 2017). A maximum hourly rain intensity of 40 $\text{mm}\,\text{h}^{-1}$ was recorded between 09:15 UTC and 10:15 UTC (i.e. 40 mm in 1 hour is exceeded once every 50 years).

Parts of the highways around Amsterdam were temporarily closed for traffic due to the rainfall. Throughout the city floodings
were reported, mostly in the centrally located neighbourhoods Oud-West and Oud-Zuid (see Fig. 2). Areas for the survey were based on a density analysis of fire brigade and municipal flood data of the City of Amsterdam.

The case study area is characterised by multi-family houses (i.e. apartment buildings) built in the period of 1880–1940 and mostly connected to separate sewer systems. The percentage of impervious surface areas is 61%, based on 2016 GIS data provided by the City of Amsterdam. The area is known for having many semi-basements (i.e. souterrains) which are vulnerable
to flooding; an exact number on the percentage of houses with a basement could not be obtained from public data sources. The case study area is practically flat (height differences of 2–3 meters). Besides pluvial flooding, we investigated cases of roof leakages in this case study, too. This survey included not only data from the 28 July 2014 event, but also other rain event that occurred after 2010.



### 2.1.2 Münster

The city of Münster (population: 310,000, area: 300 km$^2$) and the smaller town Greven (population: 37,000 area: 140 km$^2$) were also hit by extreme rainfall on 28 July 2014. The event, which exceeded a return period of 100 years, was a result of an interaction between a stationary cold front over Münster and constantly incoming hot and humid air from the east (Gruning and

Grimm, 2015). Between 14:00 UTC and 21:00 UTC, a rain intensity of 292 mm in 7 hours was measured at the weather station 'Hauptkläranlage', north of the city center of Münster, operated by the State Environmental Agency of North Rhine-Westphalia (LANUV NRW). At its peak, a depth of 220 mm was accumulated in 1.75 hours.

Except for the west, the whole city of Münster and all of Greven were affected by pluvial flooding. There was no flooding of a river system in that region that day. More than 7,000 residential houses were damaged and around 24,000 households were

without electricity for some hours. The rail and road traffic was disrupted that day. The total damage to private households for Münster is estimated to be more than EUR 70 million (GDV, 2015). The most affected neighbourhoods in Münster were located in the east of the city.

Ground elevation differences in Münster are up to 30–60 meters. The percentage of impervious surfaces in the city centre is around 90% and on a city-wide level 34%. Münster has a high percentage of single-family houses, built in the period of 1950–

1990. There is an intensive residential use of souterrains by students. Around 80% of the city area has separate sewer systems (Gruning and Grimm, 2015). The city of Greven directly borders to the city of Münster, but is part of another administrative district (i.e. Steinfurt). Greven is a small mid-sized town, with mostly small single-family houses; the earliest dating back to the 19$^{\text{th}}$ century.

The case study area compromises neighbourhoods in Münster and Greven that were most affected (Fig. 2), based on fire

brigade data on street level provided by cities the of Münster and Greven. All streets were selected, that had at least one, for Münster, or three, for Greven, fire brigade records on 28 July 2014. This case study focuses on households that suffered from pluvial flooding, which was the scope of the EVUS project that funded the Münster survey. In the remainder of the paper, we refer to 'Münster and Greven' as 'Münster'.

### 2.2 Damage data collection procedure

To identify factors that influence damage and gain insights on coping strategies, we conducted surveys among tenants and homeowners in Münster and Amsterdam whose houses were flooded due to rainfall. In line with the work by Van Ootegem et al. (2015), the surveys were also applied to flooded households that did not suffer any damage. The member of the household with the best knowledge of the damaging event was asked to participate in the survey. We aimed for a minimum of 300 completed interviews per case study to avoid small sub-samples (e.g. groups of respondents that take a certain precautionary measure).

A questionnaire was developed for the collection of damage data associated with extreme rainfall events, building upon an existing questionnaire for fluvial flooding (Thieken et al., 2005; Kreibich et al., 2005). River or groundwater flooding are not addressed in this questionnaire. The questionnaire was organised in six thematic groups, containing 82 mainly closed questions. The questionnaire acquires information on financial losses caused by damage to building structure and content,





hazard and building characteristics, people's precautionary behaviour and emergency response. A more detailed description of the questionnaire design is given in Appendix A.

In Amsterdam, we conducted computer-aided telephone interviews (CATI) and an online survey. Samples were randomly drawn from a database of landline and cell-phone numbers (2269 households) held by EDM, a customer data analytics company, for the selected case study area. A team of trained students carried out the CATI in the period of 20 January 2016 to 28 April 2016. We conducted an online survey among 7000 households for which we were not able to retrieve a phone number. Survey participants who suffered damage from multiple rain events were asked to focus on the most recent event. For Amsterdam, the entire database of survey responses is available under Creative Commons Attribution-NonCommercial license (CC BY-NC) and can be downloaded from the DANS archive (Spekkers, 2016).

In Münster, a CATI among tenants and homeowners was conducted by *explorare*, an independent market research institute. Samples were drawn from the Deutsche Post Address database (7445 households) for the affected streets. The generic questionnaire was adapted for this case study to be consistent with existing flood damage databases. More details on the survey modes of the Amsterdam and Münster case studies and the sampling procedures are given in Appendix B.

Some post-processing activities were performed on the collected data. Checks were performed to correct or remove implausible inputs, e.g. by comparing reported water levels inside and outside the house and by comparing reported floor areas with building footprint. Responses to open questions (e.g. the 'Other' field of the question 'How did water get into your house?') were manually categorized. First, open answers were categorized using existing answer categories wherever possible. If the open answer did not fit in any of existing categories, but was given by several respondents, a new category was added. Otherwise the answer was set to 'Other'.

## 2.3 Data analyses

Table 2 presents an overview of the collected data used for analyses in this paper. Similar to the paper by Thieken et al. (2007); Kienzler et al. (2015); Rözer et al. (2016), the risk management cycle (Fig. 1) is used as a framework for the data analyses and the presentation of the results. In the present study we did not cover the topic of recovery, because this would require repeated surveys over a period of time.

*Response* is here defined as the efforts to minimize the damage created by a disaster by taking emergency measures just before, during or immediately after the event. This topic covers items labelled 'Response' in Table 2. People's response was analysed by means of a frequency analysis of the emergency measures people took. A few emergency measures were only asked in one of the two case studies. In the present paper, we only report on emergency measures that were considered in both case studies.

*Risk analysis and event assessment*, in this paper, relates to the analysis of damage characteristics and the factors influencing damage. This topic covers items labelled 'Risk analysis' in Table 2. We distinguished between damage to building structure and building content, as well as the total damage. Building structure is here defined as everything permanently connected to the building, such as building walls and ceiling, permanent flooring and infrastructure. Building content are portable goods and semi-permanent objects, such as furnishing, curtains and carpets. Total damage was calculated by summing building structure



damage and building content damage for the records were both values are available including reported zero values. We analysed the effect of the following binary variables on damage:

- water contamination by sewage, chemicals, oil or gas;

- presence of a basement;

– if respondent was at home;

- respondent's awareness of the upcoming severe weather event;

- respondent's experience with water intrusion;

- if respondent took at least one precautionary measure.

We performed a median ratio test to analyse the significance of these variables, i.e. by comparing the median damage in
the subset of the data for which the binary variable is *true* with the median damage in the subset of the data for which the binary variable is *false*. For this purpose, we estimated the confidence intervals of the difference between the medians using a bootstrapping method with 10,000 bootstrap samples (e.g. Haukoos and Lewis, 2005).

*Disaster risk reduction* is here defined as a set of actions that is taken as precautionary measures in the face of a potential disaster and refers to items labelled 'Disaster risk reduction' in Table 2. We investigated the number and the type of precau-
tionary measures respondents took, as well as when respondents implemented these measures. A few precautionary measures were excluded from the analysis because they were only investigated in one of the two case studies. The correlation between people's preparedness and their experience with previous damaging rain events was determined by comparing the mean number of precautionary measures people have taken before the event in groups of respondents with and without previous flood experience. Experience is here defined as having at least one experience with a damaging rain event, independent of the severity
and the recency of earlier events. A two-sided t-test was performed to test if means are significantly different.

## 3 Results

### 3.1 Summary statistics of the data set

A total of 859 questionnaires were completed, including 510 for Münster and 349 for Amsterdam. Basic statistics are summarised in Table 3. The response rate was computed by dividing the number of completed questionnaires by the number of
contacted households times 100. In Amsterdam, the response to the CATI survey (9.3%) was higher than the online survey (2.0%). The CATI survey of Münster was in between with a response rate of 6.9%. In the CATI survey, multiple call attempts were made to obtain a completed questionnaire, whereas for the online survey we only sent out a survey invitation letter once. The interviews averaged eight minutes longer in Münster than Amsterdam mainly because of a difference in the length of the questionnaires.



Response bias was checked by comparing demographic indicators between response sample and census averages. Respondents in both cities are relatively old, highly educated and more often homeowners, compared to city level averages (Table 3). There can be several explanations for this. In the Münster survey, only landlines phone numbers were available. Due to the increasing use of cell-phones, elderly people may tend to be overrepresented in a landline only sample, as argued by Kienzler et al. (2015). In the Amsterdam survey, selected areas affected by flooding were more expensive and the sample is therefore not representative for the city as a whole.

Unpublished research by the second author, based on data from a previous study (Rözer et al., 2016), shows that demographic variables, similar to those listed in Table 3, do not correlate with damage. The exception is the variable 'Percentage of homewoners', which shows a weak positive correlation with damage. We therefore expect that damage amounts reported in this study may be overestimated because of the response bias. More details on a possible response bias are given in Sect. 4.

### 3.2 Frequency analysis of emergency response data

39% of the respondents in Amsterdam and 71% of the respondents in Münster have implemented at least one emergency measure before or during the event out of the 11 emergency measures compared in this study. Compared to similar studies by Rözer et al. (2016) for pluvial floods and Kienzler et al. (2015) for fluvial floods, the percentage for Münster is high and the percentage for Amsterdam among the lowest. However, these results should be interpreted with caution as a large number of respondents in Amsterdam did not answer this question (43–45%; see also Sect. 4.1.2).

Figure 3 shows an overview of the implemented emergency measures in the two cities. For 8 out of 11 emergency measures, the percentage of respondents who implemented emergency measures is significantly higher in Münster than in Amsterdam on a 0.001 significance level. 'Pumping or mopping out the water' is in both cities by far the most frequently implemented measure (Münster 52%, Amsterdam 23%). The measure 'Moving furniture to higher floors' ranks second in Münster (37%) and third in Amsterdam (12%). These findings are in line with studies by Rözer et al. (2016); Kienzler et al. (2015), where the two above-mentioned emergency measures are also among the three most frequently implemented measures. A survey among pluvial flood affected households in Flanders, Belgium, revealed a similar percentage for 'Moving furniture to higher floors' as in Amsterdam (Van Ootegem et al., 2015).

Unlike the measures 'Pumping or mopping out the water' and 'Moving furniture to higher floors', other measures differ considerably in popularity between the two cities. For example, the measure 'Provisionally sealing openings' ranked second in Amsterdam (13%), but was one of the least popular in Münster (18%). The differences in emergency response can partly be explained by the differences in event magnitude. Some measures are more sensible to take than others depending on the flood depth, as is discussed in more detail in Sect. 4.2.2.

### 3.3 Risk analysis and event assessment

67% of the respondents in Münster and 84% of the respondents in Amsterdam reported on structural damage to the building they live in, which includes reports of zero damage (Table 4). 64% of respondents in Münster and 93% in Amsterdam could state their damage to building contents. In the Amsterdam sample people reported a high number of zero losses for content





damage (215 out of 325 records). A similar result was found by Van Ootegem et al. (2015) who argue that these zero damages stem from the fact that "it is possible that people are able to remove the water immediately before/during the flood or they are able to protect their belongings in some way (for instance by moving them to another place)". The number of zero values is limited in the Münster sample. Since water depths in Münster were a few decimetres high, which suggests that people were

not able to remove water or protect their contents effectively.

Figure 4 shows the distribution of the total damage (top panel), the building structure damage (middle panel) and building content damage (bottom panel). There is a large variation in the loss amounts reported by respondents, ranging from tens of euros to hundreds of thousands of euros. Based on a comparison of the medians of the distributions, significantly higher amounts were observed in Münster than in Amsterdam; the median of the total damage is an order of magnitude larger in

Münster (EUR 10500) than in Amsterdam (EUR 1200). A possible cause that can explain these differences is the difference in reported water depths between the cities (Table 5), which is discussed in Sect. 4.2.3.

The damage distributions of Münster, especially for structural building damage, are less symmetrical than those of Amsterdam and show higher peak densities. This is partly because Greven and Münster are two cities with different exposures to extreme rainfall. On closer inspection of the subsets of data related to Greven and Münster, it was found that in Münster

systematically higher damage amounts were reported compared to Greven, for the same rainfall event. Figure 4 is in fact a joint density function of the density functions of Greven and Münster. But this joint effect could only partly explain the asymmetry. The asymmetry may indicate the presence of atypical extreme observations, as discussed in more detail in Sect. 4.2.3.

Figure 5 shows pathways for rainwater entering buildings as reported by respondents. In Münster, 83% of the total damage was caused by water entering the house through toilets, sinks, drains, basement entrances, doors and other openings at ground

level. In Amsterdam, only 39% of the total damage was associated with these pathways. This is partly explained by the fact that respondents who reported roof leakage not actively approached in the Mïnster survey. In Amsterdam 19% of the total damage was caused by leaking roofs. The remaining difference is probably caused by the difference in the severity of the two events (see Table 1), combined with differences in building topology between cities, but this hypothesis could not be tested based on the available data.

A number of explanatory variables for damage were investigated in this study (Fig. 6). For Münster, we found a significant difference between respondents who reported contaminated water and those who did not, in terms on median damage. Contaminated flood water positively correlated with the median damage. No significant correlation was found for Amsterdam because the number of respondents reporting contaminated water was low (Table 5). In Amsterdam, the presence of a basement significantly affected the median damage with a factor 2.2. Since less than 2% of the respondents in Münster did not report a

basement, more data are needed to be conclusive about the significance of this variable for this city. No significant correlations were found between median damage and the variables 'Experience with water intrusion' and 'Took precautionary measures'. Awareness correlates positively with median damage for Münster. More research is needed to study the causality of these relationships.



## 3.4 Disaster risk reduction

Significantly more respondents took precautionary measures in Münster compared to Amsterdam (Fig. 7). For example, the measure 'Installing a flood water pump', is taken around six times more frequently in Münster than in Amsterdam. The exception is the measure 'Adapting the building structure', which is taken more frequently in Amsterdam. This is because in Amsterdam, unlike Münster, we also investigated roof leakages and improvements to the roof were considered building adaptation.

The list of the five most popular precautionary measures of both case studies contain the same measures, but not in same order: 'Requesting information about precautionary measures', 'Installing a flood water pump', 'Avoid expensive furnishing on the floor at risk', 'Store low-value goods on floor at risk' and 'Adapting the building structure' are frequently reported by respondents in the both cities. Apart from 'Adapting the building structure' these are measures that can be implemented at relatively low or medium costs (Rözer et al., 2016).

Results show that respondents' actions were mostly reactive: many respondents implemented precautionary measures after the event. An exception is the measure 'Installing a water pump'. The reactive approach is also confirmed by Fig. 8, which shows that respondents who have experienced water intrusion before take 1.5 to 1.7 times more precautionary measures than respondents with no experience. This is in line with studies by Kreibich et al. (2005); Bubeck et al. (2012); Kienzler et al. (2015).

Figure 8 also shows that the relative increase in uptake of precautionary measures between groups with and without experience with water intrusion seem to be independent from the number of measures implemented, as well as from the fraction of households with experience. While the majority of households in the Amsterdam dataset had experience with water intrusion (83%), the number of implemented measures was relative low with less than one measure on average per household. In Münster, only 21% of the households stated to have experience with water intrusion, but implemented on average 2.3 measures. In Sect. 4.2.1 we discuss possible explanation for the difference between cities in uptake of precautionary measures.

## 4 Discussion and recommendations

The results shown in Sect. 3 reveal considerable differences between the two cities in terms of emergency response (Fig. 3), financial losses (Fig. 4) and people's level of precaution (Fig. 7), with generally higher losses and uptake of measures in Münster compared to Amsterdam. There are several underlying effects that may cause variations. These include methodological biases as well as differences in case study characteristics, i.e. differences in the magnitude of the events in terms of rainfall intensity and recorded water depth and regional effects such as differences in the socio-economics and building topology (Table 3). In this section, the observed differences are critically evaluated in terms of possible methodological biases and differences between case studies, to derive more universal coherences. Moreover, we make recommendations for future surveys on the topic of damage data collection.





## 4.1 Methodological biases

As described in Sect. 2.2, the Amsterdam and Münster surveys are based on one generic questionnaire, which was adapted independently to the case studies. The main differences between the two surveys are related to the survey delivery mode and questionnaire structure.

### 4.1.1 Survey delivery mode

In Münster a single-mode CATI survey was conducted, while in Amsterdam a combination of a CATI and an online survey was used. Although there are many studies investigating survey mode effects, i.e. the possible sources of differences in survey outcomes such as selection bias, the effect of a particular mode on the survey outcome is not yet fully understood (Couper, 2011).

Demographics of respondent groups can be compared between the samples of different survey modes to check if the choice of the survey mode has affected the representativeness of the sample (Link and Mokdad, 2006). For Amsterdam, we found only minor differences in the demographics between CATI and online survey with a similar over-representation of older and higher educated respondents compared to census data, as shown in Table 3. We therefore conclude that the choice of survey mode does not influence population representation in the samples, i.e. Münster CATI sample, Amsterdam CATI sample and Amsterdam online sample.

The bias towards older and higher educated respondents could not have been avoided by the choice of survey mode. This bias is particularly large for the Münster sample where only landline phones were contacted. Response bias in surveys that are based on landline samples only are a well-known challenge in modern survey research. Dillman (2011) argues that because of the decreasing numbers of landline phones and accessibility to online surveys (i.e. internet connection), it becomes increasingly difficult to obtain a representative sample using a single-mode survey. A combination of a CATI with landline and cell-phone numbers and an online survey, probably brought the mean age of the respondents in Amsterdam closer to census data. Because of the small differences between the telephone and online samples in Amsterdam, we assume that by including cell-phone numbers in the sample we improved the survey coverage.

### 4.1.2 Questionnaire structure

Modifications to the questionnaire structure (i.e. wording, sequencing, response format) can significantly bias survey outcomes (Couper, 2011; Bergman et al., 1994; Porst, 2014, e.g.). In the context of the present study, an important difference between the surveys (i.e. Münster CATI, Amsterdam CATI and Amsterdam online survey) is the response format of questions related to precautionary and emergency measures. These items were designed as closed questions in the Münster CATI and the Amsterdam online survey, i.e. each measure was individually asked to the respondent. In the Amsterdam CATI, a semi-closed format was chosen. While testing the Amsterdam CATI, test respondents reported to have difficulties with focussing on closed questions that contain many sub-items, which was particularly the case with the question on precautionary measures (18 sub-items). We therefore decided to group similar kind of precautionary measures (in groups of around 3-4 items) and asked first a





closed question about whether they took measures of this class. Then clarifying questions were asked to make sure the correct precautionary measures within the group were selected. In case of doubt, the interviewer explicitly went through all items and double-checked with the respondent. However, after analysing the collected data, we found that the closed question format in the online survey resulted in a significantly higher percentage of respondents who stated to have implemented one or more

precautionary measures compared to the semi-closed format in the CATI (online survey: 34%; CATI: 15%; $p < 0.001$). The same is true for the average number of implemented measures (online survey: 0.6; CATI 0.2; $p < 0.001$). Still, these values are much smaller than the values found for the Münster survey, i.e. 64% of the respondents implemented one or more precautionary measures with an average of 2.3 measures. We can therefore conclude that besides the evident methodological bias, the level of private precaution is considerably higher in Münster compared to Amsterdam.

For the question items on emergency measures in Amsterdam, where we used a closed response format in both the CATI and the online survey, we did not find a significant difference between the two samples in terms of emergency response. However, considerably more respondents in Amsterdam did not answer to this question (online survey: 45%; CATI: 43%) compared to Münster (0.4%). This was probably caused by the fact that in Amsterdam we coded a filter question (i.e. 'Did you or another person in your household take any emergency measures as an immediate reaction to the rain event?') that allowed respondents

to skip the question on emergency measures in case they did not implement any emergency measure or had no information about it. We presume that people were unfamiliar with the term 'emergency measures' (or its Dutch translation 'noodmaatregelen') and therefore skipped the question ('No answer') or answered 'No' because the emergency measure(s) they applied where not perceived as such. Because of the high number of missing values the absolute differences between the case studies should be interpreted with caution, but we can still compare the ranks of emergency measures, which will be discussed in the next

section. Possible solutions to avoid missing values for this question in a future survey are given in Sect. 4.3.

## 4.2 Results associated with hazard and regional characteristics

Taking into account the methodological biases as discussed in Sect. 4.1, differences in the results between Amsterdam and Münster are also caused by differences in hazard and regional characteristics of the case studies. It is necessary to determine to what extent these hazard and regional characteristics play a role to better understand the factors that contribute to damage due

to extreme rainfall.

### 4.2.1 Causes of differences in precautionary behaviour

Respondents in Münster implemented more precautionary measures compared to respondents in Amsterdam (Fig. 7). This cannot be explained by the magnitude of the studied event, because there was a high uptake of precautionary measures in Münster before the event as well as after. Another explanation is the relation we found between the mean number of precautionary

measures and flood experience (Fig. 8), which was also found by other researchers (Kreibich et al., 2005; Bubeck et al., 2012; Kienzler et al., 2015), but this cannot explain the absolute difference in precaution between the cities, because flood experience results in 1.5 to 1.7 times more precautionary measures, while the mean number of implemented precautionary measures was about one magnitude higher in Münster compared to Amsterdam (see Sect. 3.4).





The absolute difference in uptake of precautionary measures may be caused by differences in how people in Amsterdam and Münster perceive risk. Based on a study in Switzerland, Siegrist and Gutscher (2006) found that in regions where people overestimate risk, some people show more preventive behaviour. They found that people's flood experiences are positively correlated to their flood risk perceptions. That means that the lack of experience with water intrusion in Münster (21% of the

respondents had experience compared to 83% in Amsterdam) may have lead to an overestimation of risk and therefore a higher uptake of precautionary measures. We recommend to include question items on risk perception in a future survey as it may explain the level of precaution, and thus also indirectly damage.

### 4.2.2 Causes of differences in emergency response

The difference in emergency response between the case studies can to some extent be explained by the magnitude of the event

in terms of reported water depths (Table 5). If we compare the rankings of the emergency measures between the two case studies, we can conclude the following. The most popular emergency measures were implemented in both cases (i.e. 'Pumping or mopping out the water' and 'Moving furniture to higher floors') and, thus, are implemented irrespectively of the water depth. Other measures were mostly applied in case of large water depths (i.e. 'Switching off gas and electricity') or in case of small water depths (i.e. 'Provisionally sealing openings'). Thus, the relative small water depths in Amsterdam not only reduced the

overall necessity of taking emergency actions, they also make some measures more sensible to take than others. Rözer et al. (2016) found a similar effect: in case studies with small water depths, people focus more on emergency measures that have the goal to keep the water out (e.g. sealing openings), rather than reducing the damage after water has already entered the building (e.g. securing or moving semi-permanent facilities).

### 4.2.3 Causes of differences in financial losses

Significantly larger damage amounts were reported in Münster compared to Amsterdam, as shown in Fig. 4. With only two case studies, it is difficult to quantify the factors that explain the variability of damage between case studies. Still, possible factors can be discussed on a qualitative level. Following the conceptual model for building damage proposed by Thieken et al. (2005), we can roughly distinguish between variables that relate to the impact to the structure (i.e. hydrological load and contamination) and the resistance of the structure (i.e. permanent resistance and temporal resistance). We expect that

the Amsterdam and Münster case were mostly different because of the impacts to structures. The hydrological load in terms of water depths was much larger in Münster than in Amsterdam. Although there are differences in building types between cities (Table 3), we believe that differences in resistance are minor or slightly in favour for Münster given the high uptake of emergency and precautionary measures (Fig. 3 and Fig. 7).

In Amsterdam the damage distribution is more symmetrical on a logarithmic scale, while the damage distribution is nega-

tively skewed for Münster. Generally, flood damage data follows a lognormal distribution (Zhai et al., 2005) and as a consequence the density function would appear symmetrical on a logarithmic scale, but in case of atypical extreme observations, standard distributions such a the lognormal are unable to capture the data well (Balasooriya and Low, 2008). The asymmetry





may indicate that the Münster sample contains some exceptional losses that are caused by different damage mechanisms than the bulk of the data. This could be a topic for further research.

### 4.3 Recommendations for rainfall damage surveys

Applying a survey in different countries or regions, as done in this study, is challenging. To make survey outcomes comparable, and thus to avoid methodological biases, surveys should to a large extent share the same response format, survey delivery mode, sampling techniques and questionnaire design (Bird, 2009). On the other hand, a survey should also be able to capture regional features, for example, in our case, country-specific building topologies, and thus it is unavoidable to introduce some differences in the set-up between surveys of different case studies.

Some of the methodological biases we encountered in our survey could have been avoided, while others are more difficult to address. For example, we sampled only landline phone numbers in the Münster CATI. Including cell-phones in the sample can increase the representativeness of the sample as shown for the case of Amsterdam and other studies (e.g. Busse and Fuchs, 2012), but this is not possible for countries where cell-phone phones are not registered at an address (i.e. in Germany). The present study also highlighted certain issues with respect to the choice of response format for some of the questions (e.g. items on precautionary measures). A helpful tool to reduce these and other methodological issues in questionnaires is to use the template proposed by Bird (2009), who listed minimum requirements on methodological details of a questionnaire to allow comparison between case studies in natural hazard sciences. Another issue relates to use of filter questions. A sparse use of filter questions can generate an unnecessarily long questionnaire that comes with fatigue effects and high drop out rates. But a wrong answer to a filter question by mistake may lead to respondents skipping a block of questions, resulting in an increased number of missing observations (see Sect. 4.1.2). Possible way to avoid this is to make use of validation questions to cross-check answers to important questions.

We recommend the use of the same IT infrastructure in all case studies, i.e. the same survey software and a shared data repository. This not only increases the comparability between studies, it also makes data analyses easier and less prone to errors. In Appendix A, the LimeSurvey-coded questionnaire used in Amsterdam is presented as an example of such an infrastructure.

### 5 Conclusions

In this paper we investigated the impacts of extreme rainfall to residential buildings in the cities of Münster and Amsterdam as well as precautionary behaviour and emergency response by households. Scientific surveys were conducted among affected residents in Münster and Amsterdam to collect information on self-reported financial losses, caused by damage to building structure and building content, as well as factors influencing damage, such as hazard, building and socio-economic characteristics. The paper presents an open source, flexible questionnaire tool that is specific to the impacts of intense local rainfall events and can easily be adapted to international case studies.

A total of 510 questionnaires in Münster and 349 in Amsterdam were completed. Reported damage varied from tens of euros to hundreds of thousands of euros. The median damage was an order of magnitude larger in Münster (EUR 10500) than





in Amsterdam (EUR 1200). The mean water depths were higher in Münster (0.49–0.57 meter) than in Amsterdam (0.05–0.16 meter). 16–22% of the respondents reported water contamination by sewage, chemicals, oil or gas.

Exploratory data analyses revealed that types of implemented emergency measures are likely to be associated with the characteristics of the hazardous event. The Münster case, with higher reported water levels than in Amsterdam, shows a preference for emergency measures to reduce damage, such as unplugging electronic devices, switching off electricity and securing semi-permanent facilities, while in Amsterdam, with only minor water levels, people responded by undertaking emergency measures to prevent damage, such provisionally sealing openings. Same types of emergency measures were preferred in both cases and are independent of the water levels: moving furniture to higher floors and pumping out the water.

The difference in magnitude of the events in Münster and Amsterdam is probably also the most important cause for the differences between the cities in terms of the suffered financial losses; in Münster significantly higher damage amounts were reported compared to Amsterdam, including some exceptionally high losses. Also the rarity of observation with no damage in Münster and the high number of zero observations in Amsterdam, shows that in Münster people were unable to prevent damage, likely due to high water levels. Within the case studies also a large variation in damage was found. Factors that are significantly associated with damage are the water contamination, the presence of a basement in the building and people's awareness of the upcoming weather event.

This study confirms the conclusions of other studies that people's previous experience with adverse events positively correlates with precautionary behaviour. However, experience cannot explain the considerably higher uptake of precautionary measures observed in Münster compared to Amsterdam. We recommend in a future survey to investigate the extent to which risk perception of extreme rainfall can explain people's precautionary behaviour.

# 6 Data availability

The databases of survey responses of the Amsterdam case are available under Creative Commons Attribution-NonCommercial license (CC BY-NC) and can be downloaded from the DANS archive (Spekkers, 2016). The questionnaire used in Amsterdam can be downloaded from the same source. The survey responses of the Münster case will be available through the HOWAS21 database (GeoForschungsZentrum GFZ, 2017) five years after the end of the EVUS project (BMBF, 03G0846B), i.e. June 2023.

## Appendix A: Questionnaire

## A1 Questionnaire design criteria

We set the following requirements prior to the development of the questionnaire:

– The main objective of the questionnaire should be to characterize damage to residential buildings as a direct results of a rain event, i.e. pluvial flooding and rainwater entering the house through roofs and facades;





– The damage assessment should distinguish between the assessment of financial damage to building structure and building content; questions related to social and physical vulnerability, such as human health, will not be part of the questionnaire as this requires a completely different questionnaire design;

– The questionnaire considers a large set of contextual variables that can potentially explain damage; this list of variables should be based on scientific literature and expert judgements;

– Definitions and variables used in the questionnaire will, as far as possible, be in line with definitions and variables used in other, related questionnaires (i.e. Kreibich et al., 2005; Thieken et al., 2005, 2007; Van Ootegem et al., 2015);

– Closed questions should be incorporated in the design as much as possible to reduce data post-processing efforts, to allow quantitative statistical analyses of the data and to allow comparison within and between data sets (Sarantakos, 2005);

– The questionnaire should be applicable to computer-aided telephone interviewing (CATI) and online surveying; to avoid a 'fatigue effect', the questionnaire should not be longer than 15–20 minutes to finish (Rathod and LaBruna, 2005);

– The questionnaire should be made generic, i.e. not specific to conditions in one country, so it can be used internationally.

## A2 Item generation

We have built upon a questionnaire developed by GFZ Potsdam and Deutsche Rück, which was originally developed to assess flood damage in the aftermath of the severe flood event that hit Germany in 2002 (Kreibich et al., 2005; Thieken et al., 2005). This questionnaire has undergone several updates since that time. It has been mainly applied to fluvial flooding, i.e. the 2002, 2005, 2006, 2010, 2011 and 2013 floods in Germany (see e.g. Thieken et al., 2007; Kienzler et al., 2015). It has also been used to investigate to pluvial flood events in Lohmar and Hersbruck in 2005 and Osnabrück in 2010 (Rözer et al., 2016). The 2010 survey in Osnabrück was part of a larger survey focusing on fluvial floods and only minor changes were made to the questionnaire. The 2005 survey in Lohmar and Hersbruck, initiated by Deutsche Rück, had a specific focus on pluvial floods and some of the original questions were tailored to this type of event without completely updating the questionnaire.

The present study is a continuation of the existing line of research. We considerably adjusted the original questionnaire in terms of question items and structure to account specifically for rainfall-related damages to residential buildings. The most important changes are the following:

– We have optimized the questionnaire from around 106 items to 82 items to increase the chance that people will complete the survey. We removed questions that were not or less relevant for extreme rainfall in cities (e.g. if people received information about river water levels or locations of dike bursts, which river was overflowing, if boulders were eroded or deposited because of high flow velocities);

– We added specific questions related to local rainfall conditions (e.g. on the causes of roof leakages, on the available drainage facilities for rainwater and if wind contributed to the occurrence of water in the house), based on findings from previous studies identifying damage explanatory factors (e.g. Spekkers et al., 2015);





- In line with the study by Van Ootegem et al. (2015), we included items to specify the amount of damage in different parts of the building (i.e. basement, ground floor), rather than asking for a total damage amount only;

- The questionnaire was translated to English and has been made more generic (i.e. not specific to Germany) to make is applicable internationally;

- The questionnaire was modified in such as way that also households with no damage could complete the questionnaire.

The new questionnaire is organised in six thematic groups. Table 6 lists the groups and example questions per group. Closed questions were used as much as possible, but were relevant respondents could select the answer items 'Other, please specify:' and 'Do not know or prefer not to say'. Question groups were sequenced in such a way that there was a smooth transition between the topics. Moreover, the group 'Hazard characteristics' and 'Building information' were put at the start of the survey
as some of the items in these groups are conditional for items in next groups. The questionnaire is programmed in the open source survey software LimeSurvey 2.05 (Schmitz, 2016). Six 'urban flooding' experts, inside and outside academia, reviewed a draft version of the questionnaire. The entire LimeSurvey questionnaire structure file (.lls) can be downloaded from the DANS archive (Spekkers, 2016).

**Appendix B: Survey mode and sampling technique**

**B1 Amsterdam**

In Amsterdam, we applied two survey modes:

1. Computer-aided telephone interviewing (CATI), where trained interviewers contacted households by phone and went through the questionnaire using a computer;

2. An online survey, where households completed a web-based version of the questionnaire.

We initially considered different survey modes, but we favoured CATI for the following reasons: 1) it is consistent with the Münster survey where a CATI campaign was planned; 2) because extreme rainfall impacts is a complex topic, a CATI approach allows for questions to be clarified where needed; 3) by phone, people could be motivated to participate to the survey even if people though there damage was not relevant for the research. But because of the high costs involved in carrying out a CATI campaign (i.e. mainly the costs of hiring staff) and the limited number of phone numbers that could be obtained for the case
study area, we eventually choose for a mixed-mode survey by combing CATI with an online survey. The online survey does not have the advantage of being able to clarify questions, which may affect the reliability of responses.

The organisation of the telephone survey included the sampling of potential survey participants, the training of a team of interviewers, setting up a call centre and call centre software and writing, mail merging and sending out survey announcement letters.



The sampling was done as follows. We listed the residential addresses located in the case study using the National Building Register (Kadaster, 2013). We only listed addresses located at ground or top floor level, because these floors are most likely to be affected by rainfall. Floor level data are not readily available in the National Building Register. We therefore created an algorithm based on house numbering logic to determine the floor per address. Per address, the phone number of the main tenant

or the homeowner were then retrieved through the data enrichment service of EDM company (www.edm.nl). EDM was able to enrich around 30% of the records with one or two phone numbers, including cell-phone numbers. According to EDM, this sample covers all groups of people in the demographic sample. Phone numbers registered in the National Do Not Call Register for consumer research (i.e. 'MOA research filter', www.moaweb.nl) were not used in this study. The sample included 44% landline and 56% cell-phone numbers.

The interviews were carried out by a team of eight MSc students of the TU Delft (four males and four females), with most of them having a background in hydrology and hydraulics. A half-day session was organized to provide the students with project background and instructions. A handbook with tips and fall back statements (i.e. standard replies to frequently asked questions by the respondents) was provided to the students. The first author was closely involved in the first weeks of the data collection phase to support the interviewers. A dedicated room with computers, phones and headsets was made available by the Product

Evaluation Lab (PEL) at the TU Delft. The call centre was available from 15:00 UTC to 21:00 UTC on weekdays in the period of 20 January 2016 to 28 April 2016. We wrote a simple web interface to manage phone calls and appointments using the R shiny package (Chang et al., 2015). Up to five calls were allowed to obtain a completed questionnaire.

A letter was sent to the households to announce the survey the weeks before they being called. A cover letter can increase people's motivation to participate to a survey. In the letter, we introduced the TU Delft, we explained the research background,

the scientific and social relevance, why we selected the participant and the Dutch privacy protection regulations the research was bounded to. We also indicated that the survey would take approximately 20 minutes to finish. People had the opportunity to opt-out if they did not feel like being called. The letters were sent in six batches during the study period to ensure there was not too much time between the letter and the call attempt. More general announcement were done through social media and local website. The city authorities of Amsterdam were informed prior to the survey.

Households for which no phone number could be retrieved through the EDM data enrichment service were sent a survey invitation letter for the online survey by regular mail. The letter contained an URL to the survey website and a unique token to open the web-based questionnaire. Compared to the telephone survey, some items were removed in the online survey to make the survey 5 minutes shorter, and thus, more likely to be completed online. Moreover, some items had been slightly rephrased, in their expression only, for online readability.

Two new variables, i.e. 'building construction year' and 'floor area', were added to each record based on the National Building Register (Kadaster, 2013).

## B2 Münster

In Münster, the survey mode was CATI. Interviews were administered in the period 20 October 2015 – 26 November 2015 (i.e. a total of 37 days) by *explorare*, an independent market research institute. They have over 10 years of experience with





household surveys on the topic of flood damage (e.g. Kreibich et al., 2005; Thieken et al., 2007, 2016). The main reason for a CATI approach was to have a data set that was consistent with existing CATI data sets.

Samples were drawn by *explorare* from the Deutsche Post Address database (6457 phone numbers in Münster and 988 in Greven) for the entire case study area. Due to German privacy protection regulations, this database only contains landline
phone numbers of households that did not opt-out from being called for surveys.

A raw text file with the question items and relevance equations was provided to *explorare*, who then coded the questionnaire in VOXCO CATI, a commercial software for professional call centres. Prior to the actual survey, a demo version of the questionnaire was made available for verification purposes. Interviewers were professionals trained by *explorare*. Depending on the available call centre capacity, 2–10 interviewers were working at the same time. The interviewers received an one hour
introduction to the topic and the questionnaire by the second author of present paper. There was a feedback round after the first five completed surveys.

The survey was announced via a press release, which was picked up by at least six local and regional newspapers as well as local radio stations. Additionally, the survey was announced through the city websites. The city authorities of Münster and Greven were informed prior to the survey. They distributed the information via online and offline public notice boards.

*Disclaimer.* Privacy of the survey participants is guaranteed as all personal data were anonymized and kept confidential.

*Competing interests.* The authors declare no conflict of interest.

*Acknowledgements.* The data collection campaigns were supported by project 'EVUS—Real-Time Prediction of Pluvial Floods and Induced Water Contamination in Urban Areas' (BMBF, 03G0846B), the University of Potsdam and Deutsche Rückversicherung AG for the Münster survey and project 'Samen met verzekeraars naar een regenbestendige stad' and Climate-KIC project OASIS for the Amsterdam survey. We
would like to thank Agnes Tan (Product Evaluation Laboratory at TU Delft) for making use of their call centre facilities, the students that conducted the interviews for the Amsterdam case, Waternet for making available the fire brigade and municipal call data for Amsterdam, the City of Münster and Greven for making available the fire brigade data. Johan Post, Wouter Botzen, Harry van Luijtelaar, Philip Bubeck, Eljakim Koopman and Lot Locher are acknowledged for their comments on a draft version of the questionnaire. We would also like to thank Meike Müller from Deutsche Rückversicherung AG, who contributed significantly to the post-processing of the Münster survey data.





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



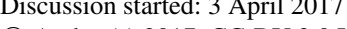

## List of Figures




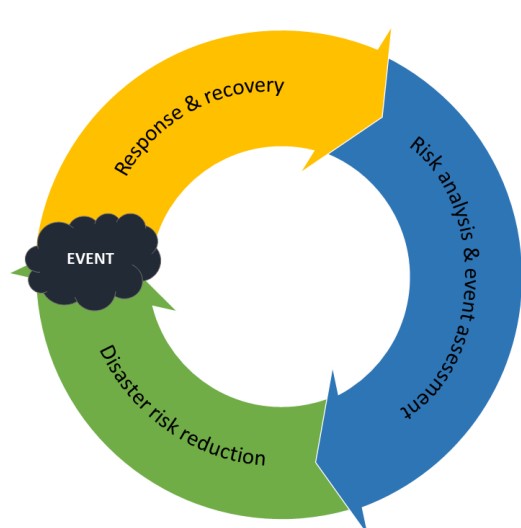

**Figure 1.** The risk management cycle used as a framework for the exploratory data analyses in this paper.


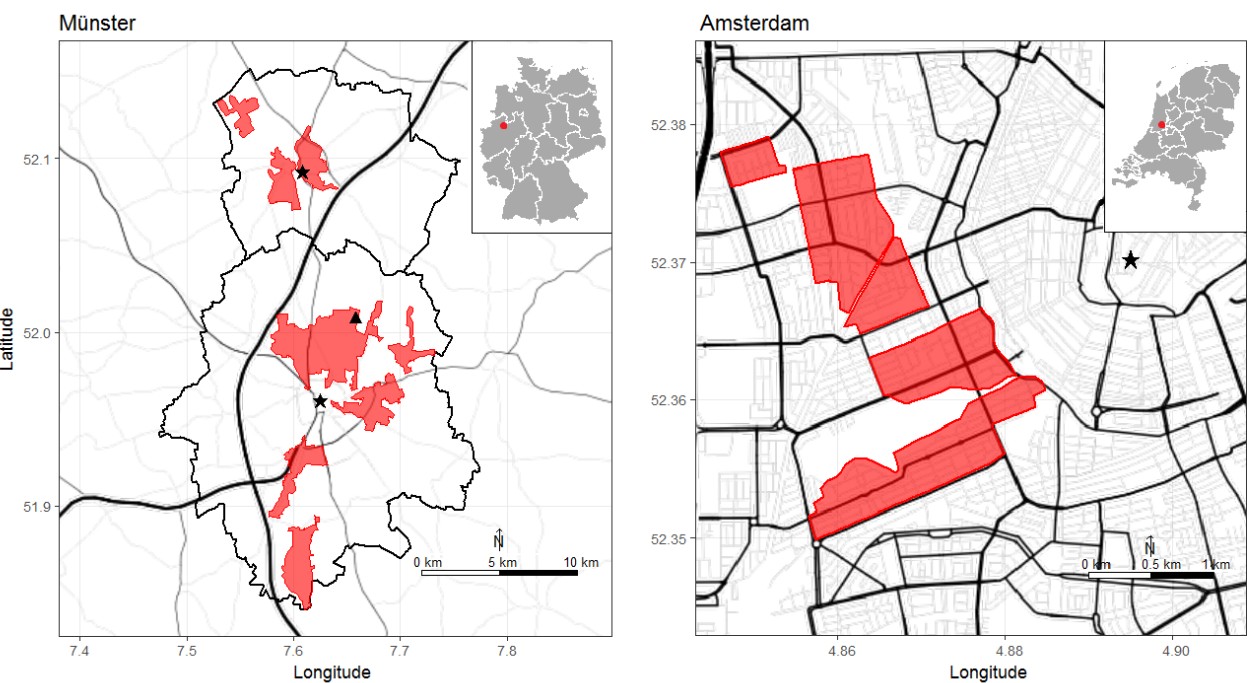

**Figure 2.** Overview map of the two case study areas. The left panel shows the cities of Münster (bottom) and Greven (top). The black triangle shows the location of the gauge 'Hauptkläranlage' in Münster. The right panel shows the neighbourhoods Oud-West and Oud-Zuid in Amsterdam. Sample areas are shown in red. The black stars indicate the centres of the three cities.



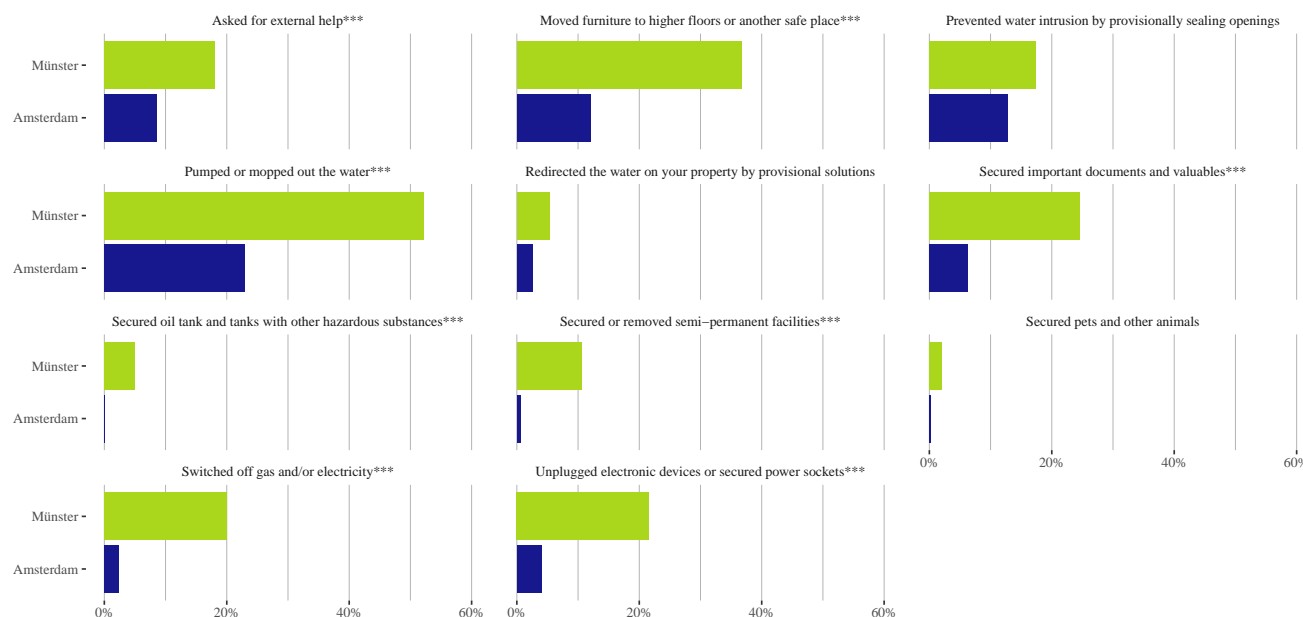

**Figure 3.** Percentage of respondents undertaking emergency measures. Only emergency measures are shown that were asked in both cities. A significant difference between proportions is denoted as follows: $* = $ p-value $< 0.05$, $** = $ p-value $< 0.01$, $*** = $ p-value $< 0.001$.







**Figure 4.** Kernel density function of the total damage (top), the building structure damage (middle) and building content damage (bottom), for Amsterdam (blue) and Münster (green). Zero values are excluded in these graphs. The vertical dashed lines represent the median of the distribution. The difference in medians ($= |x_M - x_A|$) is significant in all three plots ($p < 0.001$).




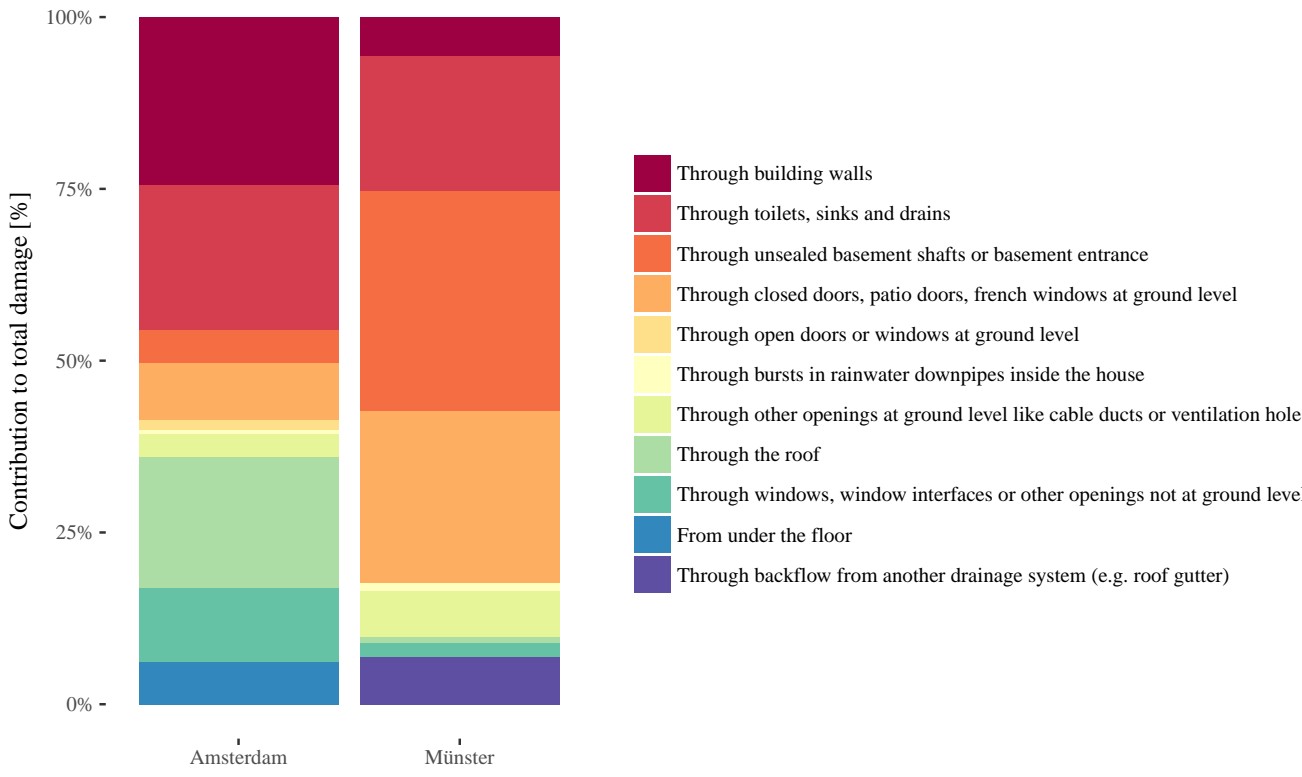

**Figure 5.** The different ways water entered a house and their relative contribution to the total damage (of all data). Damage was assigned to the pathways as follows: if a respondent only reported one pathway then the damage amount was completely assigned to that pathway. If two or more pathways were reported at the same time, then the damage amount was equally divided over these pathways.





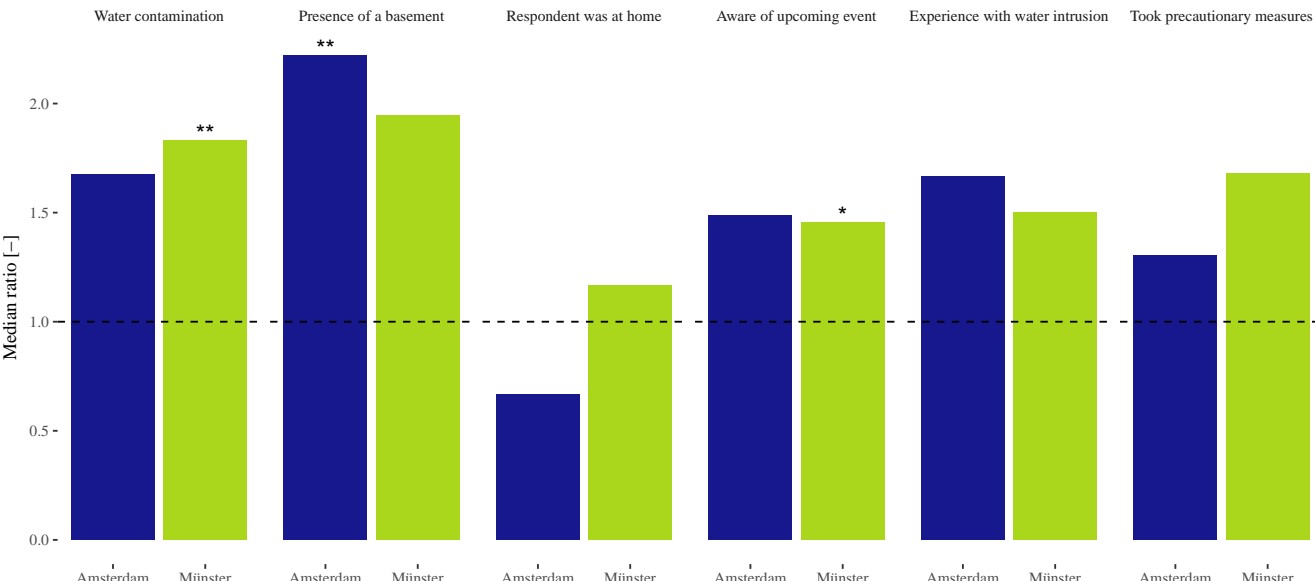

**Figure 6.** The effect of water contamination, presence of a basement in the building, presence of the respondent during the event, respondent's awareness of the upcoming rain event, experience with water intrusion and precaution on the total damage. Damage is expressed as the ratio between the median damage in the group of respondents where variable value is *true* and the median damage in the group of respondents where variable value is *false*. A median ratio above 1 means a positive correlation and below 1 means a negative correlation. A significant difference between medians, based on a bootstrapping method with 10000 bootstrap samples, is denoted as follows: $* = p < 0.05$, $** = p < 0.01$, $*** = p < 0.001$.



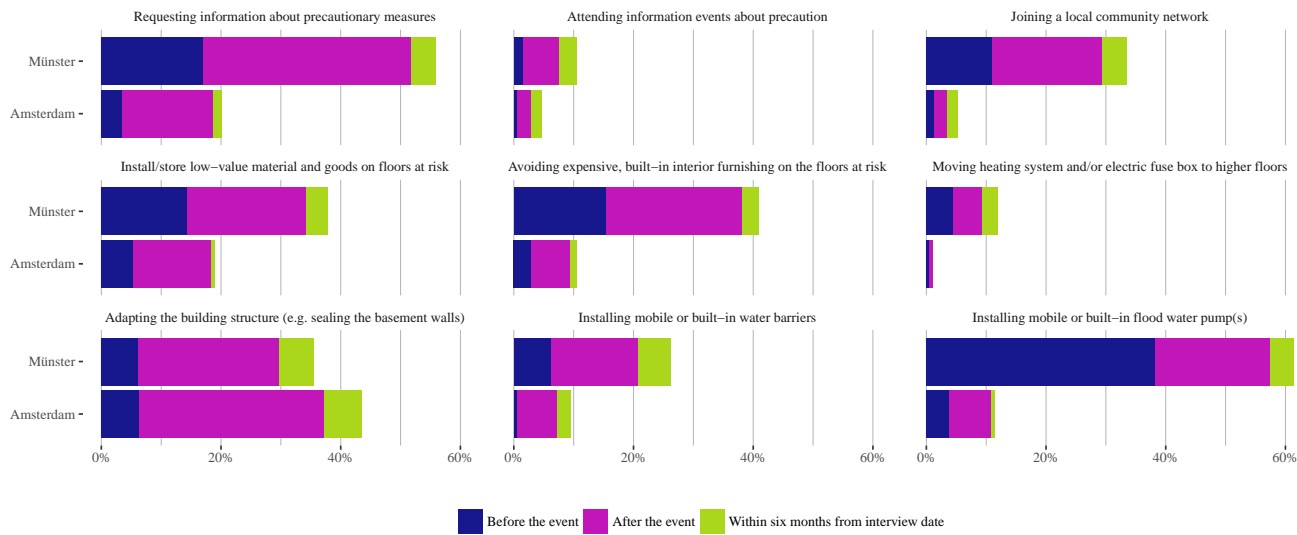

**Figure 7.** Percentage of respondents undertaking precautionary measures: before the event (blue), after the event (purple) or planned to be implemented within six months from interview date (green). Only precautionary measures are shown that were asked in both cities.




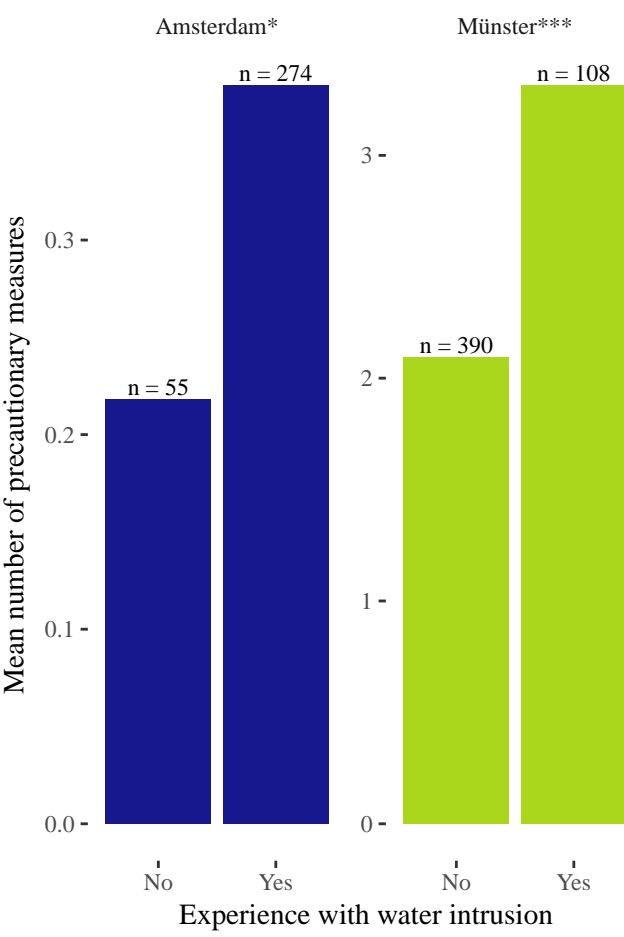

**Figure 8.** Mean number of precautionary measures against people's experience with water intrusion. A significant difference, based on two-sided t-test, between means is denoted as follows: $* = p < 0.05$, $** = p < 0.01$, $*** = p < 0.001$.





**List of Tables**





**Table 1.** Key features of the two case studies.

|  | Münster | Amsterdam |
| --- | --- | --- |
| Rainfall characteristics | 28 July 2014:<br>– 292 mm in 7 hours<br>– 220 mm in 1.75 hours | 28 July 2014:<br>– 93 mm in 6.5 hours<br>– 40 mm in 1 hour |
| Dominant building style | Single-family houses | Multi-family houses |
| Building years | 1950–1990 | 1880–1940 |
| Sewer system | 80% separate system | 75% separate system |
| Impervious surface | 34% | 61% |
| Recent flood history | No floodings before 28 July 2014 | Minor floodings |
| Survey period | 20 October 2015 – 26 November 2015 | 20 January 2016 – 28 April 2016 |
| Investigated damage processes | – Pluvial flooding | – Pluvial flooding<br>– Water intrusion through roofs |
| Survey mode | – Computer-aided telephone interviews | – Computer-aided telephone interviews<br>– Online survey |





**Table 2.** Items of the questionnaires that were used in this paper.

| Item | Measurement scale[1], unit and labels | Risk management cycle |
|---|---|---|
| **Hazard characteristics** | | |
| Water depth in basement | r: m | Risk analysis |
| Water depth at ground level | r: m | Risk analysis |
| Contaminated water | n: No \| Yes | Risk analysis |
| Entry point of water | n: How water got into the house | Risk analysis |
| **Building information** | | |
| Presence of a basement | n: No \| Yes | Risk analysis |
| Floor area | r: $m^2$ | |
| Building type | n: Detached \| Semi-detached \| Terraced \| Multi-family | |
| **Damage information** | | |
| Damage to building structure | r: EUR | Risk analysis |
| Damage to building content | r: EUR | Risk analysis |
| **Preparedness** | | |
| Flood experience | r: Number of previous flood events | Disaster risk reduction |
| Precautionary measures | n: Type of precautionary measures implemented before the event, implemented after the event and planned within six months from interview date | Disaster risk reduction |
| Aware of upcoming rain event | n: No \| Yes | Risk analysis |
| Respondent was at home | n: No \| Yes | Risk analysis |
| Emergency measures | n: Type of emergency measures implemented | Response |
| **Socio-economic variables** | | |
| Age of the respondent | r: Number of years | |
| Gender | n: Female \| Male | |
| Education | o: Highest degree of education obtained | |
| Household size | r: Number of persons living in the household | |
| Ownership structure | n: Homeowner \| Tenant | |

[1]r = ratio, o = ordinal, n = nominal.



**Table 3.** Basic statistics of the data sets. City-level census data are obtained from the databases of Federal Office of Statistics (2016) and Statistics Netherlands (2017), for Münster and Amsterdam respectively. Characteristics of people relate to persons older than 15 years.

| | Münster | | Amsterdam | | |
| --- | --- | --- | --- | --- | --- |
| | Telephone sample | Census data | Telephone sample | Online sample | Census data |
| **Survey characteristics** | | | | | |
| Number of completed questionnaires | 510 | | 210 | 139 | |
| Number of contacted households | 7445 | | 2269 | 7000 | |
| Response rate [%] | 6.9 | | 9.3 | 2.0 | |
| Mean interview time in minutes | 29 | | 21 | 21 | |
| **Demographic characteristics** | | | | | |
| Mean age of the respondent | 64 | 45 | 56 | 54 | 43 |
| Female/male ratio | 1.3 | 1.1 | 0.8 | 0.8 | 1.0 |
| Percentage of people with Master degree or higher | 37 | 20 | 50 | 55 | 38 |
| Mean household size | 2.3 | 2.2 | 2.3 | 2.3 | 1.8 |
| Mean floor area [m$^2$] | 130 | 95 | 110 | 100 | - |
| Percentage of homeowners | 80 | 42 | 66 | 63 | 39 |
| Percentage of single-family houses | 33 | 32 | 19 | 16 | - |



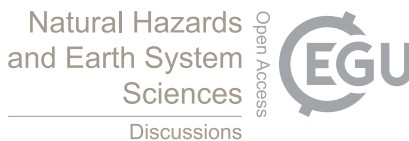

**Table 4.** Number of respondents providing loss information.

|  | Damage data | Missing values | Zero damage |
|---|---|---|---|
| Münster (n = 510) |  |  |  |
| – Structure damage | 340 (67%) | 170 (33%) | 17 |
| – Content damage | 328 (64%) | 182 (36%) | 9 |
| Amsterdam (n = 349) |  |  |  |
| – Structure damage | 294 (84%) | 55 (16%) | 91 |
| – Content damage | 325 (93%) | 24 (7%) | 215 |



**Table 5.** Reported water depths and contamination. Median and mean are based on non-zero values of the water depth.

|  | Münster | Amsterdam |
| --- | --- | --- |
| **Water depth in basement** | | |
| Median [m] | 0.35 | 0.05 |
| Mean [m] | 0.49 | 0.16 |
| **Water depth at ground level** | | |
| Median [m] | 0.20 | 0.02 |
| Mean [m] | 0.57 | 0.05 |
| Percentage of cases with contaminated water | 22 | 16 |





**Table 6.** Questionnaire: item groups and example question items

| Item group | Number of questions | Example question items |
|---|---|---|
| Hazard characteristics | 12 | – On which date did water get into your house?<br>– How did water get into your house?<br>– What was the cause of the roof leakage?<br>– Which floors of your house were affected by water?<br>– Could you give an estimate of the water depth in centimeters in the basement and on the ground floor?<br>– How long did the water remain in your house?<br>– Was the water contaminated or dirty? |
| Building information | 17 | – Which of the following building types best describes your house?<br>– Do you have a garden adjacent to your house?<br>– Which floors does your house have?<br>– What is the main flooring material being used for the following floors?<br>– Is the roof flat or pitched? |
| Damage information | 14 | – Did you have damage to your building structure and your building content, or both?<br>– Have there been any deformations or collapses of walls or ceilings?<br>– What is the total amount of building structure damage in euros?<br>– Which building contents were lost or had to be replaced after the rain event?<br>– Could you still live in your house? |
| Preparedness | 21 | – Were you or someone else at home at the time of the rain event?<br>– Were you aware of the rainstorm just before it occurred?<br>– Which emergency measures were taken as an immediate reaction to the rain event?<br>– How many times have you experienced rainwater intrusion in your life before?<br>– Have you taken any actions to store rainwater in your garden or improve the infiltration capacity of your garden? |
| Damage compensation | 8 | – Have your received any form of financial compensation from a third party?<br>– What was the size of the insurance claim in euros?<br>– How much compensation have you received by your insurer so far? |
| Socioe-conomic variables | 10 | – Do you or someone else in your household renting or owning the house?<br>– How many persons are permanently living in your household?<br>– What is the net household income per year?<br>– What is the highest education you have achieved? |
| Total number of questions | 82 | |