# Peer review of "A comparative survey of the impacts of extreme rainfall in two international case studies"

_Natural Hazards and Earth System Sciences, 2017_

## Referee Comment (RC1) · Anonymous Referee #1 · 3 May 2017

General comment:

This paper makes a useful contribution to the need for assessing the damage potential of extreme rainfalls events and the determinants of precautionary behaviour and emergency actions of private households. This work also moves beyond typical pluvial flooding by including damages that are not caused by surface flooding (e.g. rainwater directly entering the building through roofs). The methods used for data collection and analyses appear justified and appropriate. Overall, the paper is well-written, well-structured and the presented results are supported by the analyses. Though the article is scientifically sound and worth to be published, there is a number of comments and recommendations to be considered, as outlined in the "specific comments" below.

–

[Figure]

Specific comments:

Are the numbers for building style, years, sewer system also based on the 2016 GIS data (see Table 1.)? Please clarify from which source these figures were taken. The survey in Amsterdam does not only include data from the 28 July 2014 event, but also from other rain events that occurred after 2010. What does this mean and how does this affect the results? Please expand on this. Please provide a source for the data presented in paragraphs 2 and 3, p.5 Please provide more information on the composition of the Münster sample: % share of Münster, % share of Greven? It would be useful to provide information on how robust the results are, when the Münster sample would only include data from Münster, rather than lumping Münster and Greven into one and the same sample. What do you mean by "sample people" in last paragraph, p.8. Is this a typo? I believe it is not necessary to explain how the response rate was calculated (p.7). Presenting the actual response rates is sufficient. Please try to stick always to the same order when mentioning Münster and Amsterdam in the text. Also, keep the order of the two cities consistent across figures and tables. An inconsistent use of the order of the cities throughout the manuscript, tables and figures unnecessarily confuses the reader. paragraph 3, p.8: Please note how missing values were treated. Were missing values included in the calculation of the 39 % in Amsterdam? Without providing information on missing data, the figures are difficult to interpret. Wording regarding p-values should be used consistently: see "$p < 0.001$" in Figure 4 and "p-value $< 0.001$" in Figure 3. Please expand on or clarify "This is partly explained by the fact that respondents who reported ..." line 20, p.9. Also, there is a typo in this sentence (the ü is missing in Münster). last paragraph, p.11: "presented" instead of "asked" would perhaps improve the wording. first paragraph, p.17: "to make is applicable" should probably read "to make it applicable". second paragraph, p.17: "much as possible, but were relevant" should probably read "much as possible, but where relevant" second paragraph, p.17: use past tense in line 10, as in the rest of the paragraph ("was" instead of "is"). paragraph 1, p.13: There seems to be a problem regarding the causal chain "high flood experience" leads to "high risk perception" leading

to "preventive behaviour". If this argument was true, households in Amsterdam (more flood experience) would have taken more precautionary measures than households in Münster (less flood experience). In fact, it seems the opposite is true. The argument that a lack of experience with water intrusion may have lad to an overestimation of risk does not seem to be plausible in this case. Please reconsider or clarify the causal change used in this paragraph. Further, the relationship between risk perception and flood precautionary behaviour is contested in the literature. A significant number of studies found that these two factors are unrelated. Please re-check the literature and expand on the argument.
* * *

---

## Referee Comment (RC2) · Anonymous Referee #2 · 8 May 2017

GENERAL COMMENTS

-The overall impression of the paper is good. The paper is well-written, builds on relevant previous findings, has a clean structure and an appropriate tone.

-Overall, I think the comparison of case studies from different countries is relevant and worthwhile as it brings the science together and bridges international differences, which seems particularly important in relation to the topic of pluvial floods. Although, the comparison of the two presented case studies is challenging given the inherent inhomogeneity, the study gives relevant insights, while also providing tools and recommendations for future research in this field.

-The title might be more specific as to which impacts of extreme rainfall (i.e. impacts on residential buildings) are addressed.

-The applied methods seem appropriate and are generally well described. However, related to the self-reported financial losses, it is not quite apparent how the consistency of the estimates by tenants and homeowners was assured (see also specific comments). Is it safe to assume that each respondent estimated the damage costs using the same assumptions (e.g. replacement costs, including or excluding costs for cleaning, considerations of deductibles etc.)? Do tenants have access to the same information as homeowners, so that the loss estimates are comparable? Furthermore, are the damage values representing damage values of the whole building and if yes, how were the damages to multi-family houses considered?

SPECIFIC COMMENTS

-P2, L6-7: I do not think that it is safe to assume that generally a considerable amount of damage is caused by water entering through the roofs. Spekkers et al. (2015) have shown that this holds for pluvial damages in Rotterdam, the Netherlands, and not elsewhere. These characteristics are likely reflecting regional or national characteristics and are not of general nature.

-P2, L15-32: In these two paragraphs, two approaches to collect ex-post damage data are presented and compared. As a reader, I am under the impression that loss data from risk transfer schemes are more prone to biases than data from scientific surveys. Firstly, I would argue that the potential biases are heavily dependent on the national or regional risk transfer scheme and particularly the respective insurance scheme. Secondly, the potential biases can usually be accounted for. At the same time, similar potential biases exist for data stemming from surveys, which is not mentioned. Therefore, I think it is appropriate to stress the apparent advantage of a complete temporal and spatial picture provided by risk transfer data (mentioned), opposed to the advantage of being able to consider many different factors by survey data (mentioned). As respective disadvantages, it might be worth mentioning that data from risk transfer schemes might be difficult to obtain due to privacy reasons (not mentioned), while survey data are expensive to collect and depend highly on the willingness of the affected people to

participate in the survey (mentioned).

-P2, L31-32: It is stated that an advantage of survey data over data from risk transfer schemes is the possibility to interrogate people that have not suffered any damage during specific hazardous events. However, it is not clear if and how this advantage has been exploited in the presented case studies.

-P5, L25-29: Tenants and homeowners were interviewed, while collecting data about damages to buildings and content. However, tenants might not know as much about the damages to buildings as the respective homeowner (assuming that the homeowner is responsible for building damages), while homeowners might not have detailed information about content damage of their tenants (assuming that tenants are responsible for content damage). Is this presumption applicable and if yes, how was this taken into account? Is there a link between missing values (and/or zero damage) of tenants and building damage, as well homeowners and content damage, respectively?

-P6, L34-P7, L2: Apparently, the total damage is computed in case information about the content and building damage is available. It is not clear, however, how this affects the sample sizes. This information should be reported (e.g. in Table 4 and Figure 6).

-P15, L3-8: The paragraph's message is not clear, as the first statement (i.e. the types of implemented emergency measures are related to the event's characteristics) is rather contradictory with respect to the second statement (i.e. the same emergency measures that are independent of the event's characteristics are preferred in both case studies).

-P16, L13: A generic questionnaire should is advocated. However, it is stated earlier (P14 L6-8) that regional characteristics should be taken into account, as well.

-Table 4: It would be helpful to see the relative numbers of zero damage, as well (in percent). Maybe it would be beneficial to report the numbers by means of a stacked bar plot, instead of a table.

TECHNICAL CORRECTIONS

-P2, L23: the expression "by it own" should be revised

-P6, L21-22: punctuation should be revised

-P7, L1: where instead of were

-P9, L21: the spelling of Münster and/or the whole sentence should be revised

-P9, L26: in terms of

-P13, L23 and L25: impact on

-P13, L32: such as the

-P14, L16: relates to the use

-P14, L18: However, a wrong

-P14, L19: A possible way

-P15, L1: were a lot / much higher

-P15, L7: such as provisionally

-P15, L11: the expression "rarity of observation" should be revised

-P17, L5: such a way

-P17, L23: people thought their damage

-P20ff: Check the references' dois and omit weblinks where dois are available, e.g. P21, L27-28: omit link to discussion paper; P21, L31-33: omit (two different) web links; P22, L5-6: wrong doi

---

## Short Comment (SC1) · 26 May 2017

The authors wish to thank the editors and reviewers for their time and effort for reviewing our manuscript. We appreciate the valuable comments. They were highly insightful and enabled us to improve the quality of our manuscript. In the following pages are our point-by-point responses to each of the comments of the reviewers. Revisions in the manuscript are shown using red highlights for additions, and strikethrough font for deletions. We hope that the changes have improved the manuscript to a level that is suitable for publication, and we look forward to your response. The most substantial revisions concern improvements to the description of the survey, the resulting data set and the discussion concerning the relationship between risk perception and precautionary behavior. In addition, we revised unclear and/or ambiguous paragraphs throughout the manuscript to improve the readability.

[Figure]

We are looking forward to your decision on our article.

Reviewer 1

General comments This paper makes a useful contribution to the need for assessing the damage potential of extreme rainfalls events and the determinants of precautionary behavior and emergency actions of private households. This work also moves beyond typical pluvial flooding by including damages that are not caused by surface flooding (e.g. rainwater directly entering the building through roofs). The methods used for data collection and analyses appear justified and appropriate. Overall, the paper is well-written, well structured and the presented results are supported by the analyses. Though the article is scientifically sound and worth to be published, there is a number of comments and recommendations to be considered, as outlined in the "specific comments" below.

Response: Thank you very much for your time and effort to review this manuscript. Your suggestions were very helpful for improving the quality of the manuscript. We followed all of your suggestions and changed the relevant sections accordingly. Please see our responses to your specific comments below.

Specific comments:

Are the numbers for building style, years, sewer system also based on the 2016 GIS data (see Table 1.)? Please clarify from which source these figures were taken.

Response: We added all sources directly to Table 1 via footnotes.

The survey in Amsterdam does not only include data from the 28 July 2014 event, but also from other rain events that occurred after 2010. What does this mean and how does this affect the results? Please expand on this.

Response: As we are analyzing pluvial flood consequences on the level of individual buildings, we consider each household as an independent observation. Therefore, we do not expect any biasing effect of including data from other events in Amsterdam on

our analysis on the building level. However, we agree that aggregated results such as the damage distribution shown in Figure 4 can and should not be interpreted in the context of a single event, but of the survey data set itself. We changed the according paragraphs in the manuscript to clarify that.

P 5, L23-24: "Since the extreme rainfall event on 28 July 2014 was most often reported by respondents (41% of all cases), we refer to this event in the event description." P 6, L11-14: "In case participants suffered from a rain event after 2010 other than the one on 28 July 2014, they were asked to report on this event. Therefore, the analysis in this study do not exclusively refer to the extreme rainfall events on 28 July 2014, but impacts of extreme rainfall in general." P 9 L18: "Figure 4 shows the distribution of the total damage (top panel), the building structure damage (middle panel) and building content damage (bottom panel) of the two data sets."

Please provide a source for the data presented in paragraphs 2 and 3, p.5

Response: Sources added to references (P4, L23 & L27).

Please provide more information on the composition of the Münster sample: % share of Münster, % share of Greven? It would be useful to provide information on how robust the results are, when the Münster sample would only include data from Münster, rather than lumping Münster and Greven into one and the same sample.

Response: We investigated the robustness of our results after splitting the data set between Münster and Greven. In fact, we found that the differences between Münster and Greven are even smaller than our initial assumption based on the differences in the damage distributions and are very likely caused by lower water levels in Greven compared to Münster. Therefore, we changed the respective parts of the manuscript (P8, L3-5; P9, L21-30) accordingly. We also added information on the shares between Münster and Greven to section 3.1.

P9, L21-30:"A possible cause that can explain these differences is the difference in

reported water depths between the cities (Table 5), which is discussed in Sect. 4.2.3."

What do you mean by "sample people" in last paragraph, p.8. Is this a typo?

Response: We added a comma to make clear, that "Sample" is supposed to refer to "Amsterdam" and not to "people"(P9, L11).

I believe it is not necessary to explain how the response rate was calculated (p.7). Presenting the actual response rates is sufficient.

Response: According to the American Association for Public Opinion Research (AA-POR), there are six different ways to calculate the response rate (see AAPOR, 2015). In order to avoid confusion, we think it is helpful to refer to the calculation used. We also added the respective reference (P8, L5-6).

Please try to stick always to the same order when mentioning Münster and Amsterdam in the text. Also, keep the order of the two cities consistent across figures and tables. An inconsistent use of the order of the cities throughout the manuscript, tables and figures unnecessarily confuses the reader.

Response: Changed order to "Münster and Amsterdam" throughout the text where possible and changed the plots accordingly (Figures 4, 5, 6 & 8). We also changed the order of chapter 2.1.1 and 2.1.2 that the description of the Münster study area is mentioned before the description of Amsterdam (P4 & P5).

paragraph 3, p.8: Please note how missing values were treated. Were missing values included in the calculation of the 39 % in Amsterdam? Without providing information on missing data, the figures are difficult to interpret.

Response: We added information concerning the handling of missing data to the text (P8, L25-27): "For the frequency analysis all observations including missing data were considered. Therefore, the results have to be interpreted with caution as a large number of respondents in Amsterdam did not answer to this question (43-45%; see also Sect. 4.1.2.)."

Wording regarding p-values should be used consistently: see "p < 0.001" in Figure 4 and "p-value < 0.001" in Figure 3.

Response: Changed "p-value" to "p" in Figure 3 to be consistent with the other figures

Please expand on or clarify "This is partly explained by the fact that respondents who reported ..." line 20, p.9.

Response: Sentence rewritten for clarification (P9, L33 – P10, L3).

Also, there is a typo in this sentence (the ü is missing in Münster). last paragraph, p.11: "presented" instead of "asked" would perhaps improve the wording. first paragraph, p.17: "to make is applicable" should probably read "to make it applicable". second paragraph, p.17: "much as possible, but were relevant" should probably read "much as possible, but where relevant" second paragraph, p.17: use past tense in line 10, as in the rest of the paragraph ("was" instead of "is").

Response: Corrected all mentioned spelling/grammar mistakes.

paragraph 1, p.13: There seems to be a problem regarding the causal chain "high flood experience" leads to "high risk perception" leading to "preventive behaviour". If this argument was true, households in Amsterdam (more flood experience) would have taken more precautionary measures than households in Münster (less flood experience). In fact, it seems the opposite is true. The argument that a lack of experience with water intrusion may have lad to an overestimation of risk does not seem to be plausible in this case. Please reconsider or clarify the causal change used in this paragraph.

Response: We agree that the narrative described in the manuscript does not follow a clear causal chain. The paragraph origins from an earlier version of the manuscript and was not meant to be in the final version. We updated and rewrote the respective paragraph (P13, L15-18):

"Based on a study in Switzerland, Siegrist & Gutscher (2006) found German-speaking regions to have a significantly lower perception of flood risk compared to Frenchspeaking regions. They also found, that people in German-speaking regions under-estimated their flood risk, while people in French-speaking regions overestimated their flood risk compared to expert judgements.

Further, the relationship between risk perception and flood precautionary behaviour is contested in the literature. A significant number of studies found that these two factors are unrelated. Please re-check the literature and expand on the argument.

Response: Thanks for this important comment. We agree that the relationship be-tween risk perception and precautionary behavior is not sufficiently discussed in this paragraph. We rewrote the respective parts and added a more elaborate literature review concerning this topic (P13, L18-21):

"However, the relationship between risk perception and precautionary behaviour is sub-ject to current research and not yet well understood. While few studies found a signifi-cant correlation between risk perception and precautionary behaviour (i.e. Grothmann & Reusswig (2006)), a large number of studies could not find such a relationship (see Bubeck et al. (2012) for an overview)."

Reviewer 2

General comments -The overall impression of the paper is good. The paper is well-written, builds on relevant previous findings, has a clean structure and an appropriate tone. -Overall, I think the comparison of case studies from different countries is relevant and worthwhile as it brings the science together and bridges international differences, which seems particularly important in relation to the topic of pluvial floods. Although, the comparison of the two presented case studies is challenging given the inherent inhomogeneity, the study gives relevant insights, while also providing tools and rec-ommendations for future research in this field. -The title might be more specific as to which impacts of extreme rainfall (i.e. impacts on residential buildings) are addressed.

Response: Thank you very much for your time and effort to review this manuscript.

Your suggestions were very insighful for improving the quality of the manuscript. We followed almost all of your suggestions and changed the relevant sections accordingly. Although we considered a more specific title, we would like to keep the current title. The presented manuscript is intended for an audience, which is not only interested in the results of the survey itself but also in the questionnaire. Therefore, we would also like to attract readers who are interested in altering the questionnaire outside the scope of residential buildings.

-The applied methods seem appropriate and are generally well described. However, related to the self-reported financial losses, it is not quite apparent how the consistency of the estimates by tenants and homeowners was assured (see also specific comments). Is it safe to assume that each respondent estimated the damage costs using the same assumptions (e.g. replacement costs, including or excluding costs for cleaning, considerations of deductibles etc.)? Do tenants have access to the same information as homeowners, so that the loss estimates are comparable? Furthermore, are the damage values representing damage values of the whole building and if yes, how were the damages to multi-family houses considered?

Response: Thanks for this comment. We added information on how we treated homeowners and tenants concerning the collection of damage data. We also added more detailed information in regard to this topic to the appendix, in order to maintain the readability of the manuscript (see Specific comments for details).

Specific comments:

-P2, L6-7: I do not think that it is safe to assume that generally a considerable amount of damage is caused by water entering through the roofs. Spekkers et al. (2015) have shown that this holds for pluvial damages in Rotterdam, the Netherlands, and not elsewhere. These characteristics are likely reflecting regional or national characteristics and are not of general nature.

Response: We agree with the reviewer, that the mentioned example in Spekkers et

al. (2015) might not apply in general. However, we would like to mention that roof leakages are an often overseen aspect in the literature on damage caused by extreme rainfall. We rephrased the sentence for clarification (P2, L6):

"Damage can also be is caused by rainwater directly entering the building through roofs (Spekkers et al., 2015)."

-P2, L15-32: In these two paragraphs, two approaches to collect ex-post damage data are presented and compared. As a reader, I am under the impression that loss data from risk transfer schemes are more prone to biases than data from scientific surveys. Firstly, I would argue that the potential biases are heavily dependent on the national or regional risk transfer scheme and particularly the respective insurance scheme. Secondly, the potential biases can usually be accounted for. At the same time, similar potential biases exist for data stemming from surveys, which is not mentioned. Therefore, I think it is appropriate to stress the apparent advantage of a complete temporal and spatial picture provided by risk transfer data (mentioned), opposed to the advantage of being able to consider many different factors by survey data (mentioned). As respective disadvantages, it might be worth mentioning that data from risk transfer schemes might be difficult to obtain due to privacy reasons (not mentioned), while survey data are expensive to collect and depend highly on the willingness of the affected people to participate in the survey (mentioned).

Response: We added the mentioned points to the paragraphs on flood damage data (P2, L24-25 & L31-33):

P2, L24-25: "In addition, access to damage data from risk transfer schemes and similar sources might be constrained by data privacy protection." P2, L31-33: "Depending on the questionnaire and survey mode, this sample can be biased through an over-representation of certain groups (selection bias) or a cognitive bias caused by the questionnaire (response bias)."

-P2, L31-32: It is stated that an advantage of survey data over data from risk transfer

schemes is the possibility to interrogate people that have not suffered any damage during specific hazardous events. However, it is not clear if and how this advantage has been exploited in the presented case studies.

Response: Since the introduction is meant to frame our work in a broader context, we did not go into detail concerning our own analysis. However, the effect of so called "zero damage" cases on our analysis is discussed throughout the manuscript (see P9 L9-16; P15 L30-31 and Table 4).

-P5, L25-29: Tenants and homeowners were interviewed, while collecting data about damages to buildings and content. However, tenants might not know as much about the damages to buildings as the respective homeowner (assuming that the homeowner is responsible for building damages), while homeowners might not have detailed information about content damage of their tenants (assuming that tenants are responsible for content damage). Is this presumption applicable and if yes, how was this taken into account? Is there a link between missing values (and/or zero damage) of tenants and building damage, as well homeowners and content damage, respectively?

Response: Thank you very much for this comment. We added information on how we treated homeowners and tenants concerning the collection of damage data. We also added more detailed information in regard to this topic to the appendix, in order to maintain the readability of the manuscript (P 5, L31-33; P 16, L21-27): P 5, L31-33 : "Homeowners were asked to report on their damage to building content and building structure, while tenants were only asked to report on the latter in case they had detailed knowledge about the structural damage of the building." P 16, L21-27: "- The target group of the questionnaire are private homeowners and tenants. Homeowners are asked to report their financial damage to building structure and building content. Tenants are asked to report on the building content damage of their household and, in case they have detailed information (i.e. bills), on the damage to the structure of the building they live in. - Cases where tenants or homeowners can only report on one of the damage types, the other one is considered as missing observation. In case water

entered the building, but did not cause damage to the building content and/or building structure, the respective damage is considered to be zero."

-P6, L34-P7, L2: Apparently, the total damage is computed in case information about the content and building damage is available. It is not clear, however, how this affects the sample sizes. This information should be reported (e.g. in Table 4 and Figure 6).

Response: Added sample sizes of the total damage to Table 4 and to the caption of Figure 6.

-P15, L3-8: The paragraph's message is not clear, as the first statement (i.e. the types of implemented emergency measures are related to the event's characteristics) is rather contradictory with respect to the second statement (i.e. the same emergency measures that are independent of the event's characteristics are preferred in both case studies).

Response: We agree and rephrased the paragraph for clarification (P 15, L22): "Exploratory data analyses revealed that the types of implemented emergency measures are likely to be associated with the hazard characteristics of the event, such as the water level."

-P16, L13: A generic questionnaire should is advocated. However, it is stated earlier (P14 L6-8) that regional characteristics should be taken into account, as well.

Response: We rephrased the paragraph to point out, that we are aiming for a generic questionnaire that can be easily adapted to regional specifications when applied internationally (P17, L8).

-Table 4: It would be helpful to see the relative numbers of zero damage, as well (in percent). Maybe it would be beneficial to report the numbers by means of a stacked bar plot, instead of a table.

Response: We added the percentage of zero damage cases relative to the total amount of observations for both cases to Table 4.

Technical corrections

-P2, L23: the expression "by it own" should be revised

-P6, L21-22: punctuation should be revised

-P7, L1: where instead of were

-P9, L21: the spelling of Münster and/or the whole sentence should be revised

-P9, L26: in terms of

-P13, L23 and L25: impact on

-P13, L32: such as the

-P14, L16: relates to the use

-P14, L18: However, a wrong

-P14, L19: A possible way

-P15, L1: were a lot / much higher

-P15, L7: such as provisionally

-P15, L11: the expression "rarity of observation" should be revised

-P17, L5: such a way

-P17, L23: people thought their damage

-P20ff: Check the references' dois and omit weblinks where dois are available, e.g. P21, L27-28: omit link to discussion paper; P21, L31-33: omit (two different) web links;

P22, L5-6: wrong doi

Response: Corrected all mentioned mistakes.

[revised manuscript text omitted]

---

## Author Comment (AC2) · 2 Jun 2017

We would like to thank the editor for reviewing our manuscript. The changes follow the editor's suggestion to extend the regional focus of the literature review including studies from the US and the UK. We hope that the changes have improved the manuscript to a level that is suitable for publication, and we look forward to your response.

P. 2, L. 3-8: "However, recent pluvial flood events to urban dwellings in Europe and elsewhere have demonstrated that the adverse consequences of extreme rainfall must not be neglected. This includes large cities such as the pluvial floods in Copenhagen in July 2011 with EUR 807 million of insured losses (Garne et al., 2013) or Beijing, where a rainstorm in July 2012 caused an estimated total loss of over US $1.86 billion (Wang

et al., 2013). But also smaller cities such as the city of Hull, which suffered, among other towns in the UK, from severe pluvial flooding after a series of extreme rainstorms in 2007 (Coulthard and Frostick, 2010)."

[revised manuscript text omitted]